# Effect of Enyzmes on the Quality and Predicting Glycaemic Response of Chinese Steamed Bread

**DOI:** 10.3390/foods12020273

**Published:** 2023-01-06

**Authors:** Wenjun Liu, Margaret Brennan, Charles Brennan, Linfeng You, Dawei Tu

**Affiliations:** 1School of Environment and Resources, Chongqing Technology and Business University, Chongqing 400067, China; 2Department of Wine, Food and Molecular Biosciences, Faculty of Agriculture and Life Sciences, Lincoln University, Lincoln 7647, New Zealand; 3School of Science, RMIT University, Melbourne 3000, Australia

**Keywords:** xylanase, cellulase, α–amylase, Chinese steamed bread, in vitro glycaemic response

## Abstract

The present study investigates the individual and interactional effects of α-amylase (6 and 10 ppm), xylanase (70 and 120 ppm) and cellulase (35 and 60 ppm) on the physicochemical characteristics and nutritional quality of Chinese steamed bread (CSB) incorporated with 15% oat bran. As a result, the single enzyme can significantly improve the specific volume and texture of CSB. Compared to the single enzyme, the combined enzymes improved the specific volume of CSB up to the highest value (2.51 mL/g) and decreased the hardness to the minimum value (233.61 g) when the concentration was 6, 70 and 35 ppm. With respect to chemical and nutritional properties, the addition of single enzyme had no great changes, while the combined enzymes (6, 70 and 35 ppm) significantly (*p* < 0.05) decreased the total starch from 37.52 to 34.11% and hence increased the area under the reducing sugar release curve during 2 h in vitro digestion (AUC) from 344.61 to 371.26. Consequently, enzymes combination can significantly improve the quality of oat bran CSB whereas reduce the nutritional value of oat bran CSB.

## 1. Introduction

In recent years, consumers have developed a growing awareness surronding the link between diet and nutrition, and thus there has been an increased demand for healthier products with a consequent rise in interest in functional and nutritional items by the food industry [1]. As a functional ingredient, dietary fibre (DF) has been proven to have many health beneficial effects and a potential role for disease prevention [2]. Oat bran (OB) is a low-cost by-product produced during oat milling, and is a good source of DF. The main DF component of oat bran is β-glucan, which is a natural polymer composed of the glucose molecules joined by β-(1-3) and β-(1-4) glycoside bonds [3]. As a water-soluble fibre, β-glucan can easily form the viscous solutions, which slows the intestinal transit, delays gastric emptying and slows glucose and sterol absorption in the intestine [4,5]. Oat β-glucan has outstanding functional and nutritional properties due to its viscosity properties. According to the research of Liu et al. [6], the Chinese steamed bread (CSB) incorporated with 15% OB led to a reduction of predicted glycaemic response. However, previous research has shown that the addition of oat bran into CSB generally results in negative effects on rheological properties, baking performance and texture properties of final products, such as reducing the extensibility, increasing the dough stickiness, reducing loaf volume, darkening the crumb and increasing firmness [6,7].

To improve the quality of CSB incorporated with 15% oat bran, enzymes were used as individual and combination. A previous study by Liu et al. [8] reported that the addition of α-amylase, xylanase and cellulase into wheat bran dough can produce positive effects during breadmaking, such as improving the rheological behaviour of dough and hence the quality of final products. Therefore, these three enzymes (α-amylase, xylanase and cellulase) were used in this study. Fungal α-amylase is an enzyme derived from fungi, with widespread application in food industry. The action of amylase is to catalyse the hydrolysis of α-1, 4-glycosidic linkages into starch molecules (amylose and amylopectin), at a lower rate, maltodextrins and oligosaccharides [9]. Xylanase is a hydrolase, which can randomly attack the arabinoxylan (AX) backbone and break the glycosidic linkages in AX, result in changing the functional and physicochemical properties of AX [10]. Cellulase belongs to the glycoside hydrolase family, which can catalyse the hydrolysis of (1,4)-beta-D-glucosidic linkages in cellulose and other beta-D-glucan [11]. However, there is a paucity of reports on the effects of enzymes combination, especially regarding the combination of cellulase, xylanase and α-amylase on the predicted glycaemic response.

Thus, the aim of this study was to investigate the effect of α-amylase, xylanase and cellulase on the physicochemical and nutritional properties of CSB incorporated with 15% oat bran.

## 2. Materials and Methods

### 2.1. Ingredients

Wheat flour (Champion Flour Milling Ltd., Christchurch, New Zealand), oat bran (Goodman Fielder Ltd., Auckland, New Zealand), yeast powder and salt (Pams Products Ltd., Auckland, New Zealand) were purchased at a local supermarket. Megazyme Dietary Fibre and Total Starch analysis kits were purchased from Novozymes Australia Pty Ltd. (Novozymes, North Rocks NSW, Australia).

Three commercial enzymes were used: Fungamyl 2500 SG (2–10 ppm), Pentopan Mono BG (20–120 ppm) and Cellulast BG (10–60 ppm) (Novozymes, Australia).

### 2.2. Design of Experiment

Firstly, the effect of single enzyme use on the quality of CSB incorporated with 15% of oat bran was analysed by analysis of variance (ANOVA). According to the manufacture recommendations and previous research [8], the dosage of the Cellulast BG, Fungamyl 2500 SG and Pentopan Mono BG was added with 35 ppm, 10 ppm and 70 ppm, respectively.

Secondly, a full factorial 2^3^ design of experiments was used to investigate the effect of enzymes combinations on the quality of CSB incorporated with 15% of oat bran. Generally, there are three factors (α-amylase, xylanase and cellulase) at two levels (−1, 1) resulting in 8 different combinations of experiments and the coded values per each level of each factor are presented in Appendix A. According to the estimated coefficients (β_i_ and β_ij_), the theoretical response function (W) was calculated as following linear regression model:

W = β_0_ + β_1_A + β_2_B + β_3_C + β_12_AB + β_13_AC + β_23_BC + β_123_ABC



Factors: A—α-amylase; B—xylanase; C—cellulase; AB—α-amylase*xylanase; AC—α-amylase*cellulase; BC—xylanase*cellulase; ABC—α-amylase*xylanase*cellulase.

W—The theoretical response variable; β_0_—The global mean; β**_i_—**The regression coefficient corresponding to main factor; β_ij_ and β_ijk**—**_The regression coefficient corresponding to the interactions.

This multiple linear regression model with three independent variables describes the bread quality is related to the α-amylase, xylanase and cellulase.

### 2.3. Preparation of Chinese Steamed Bread (CSB)

Chinese steamed bread loaves were produced using the formulation of Liu et al. [6]. The recipe consisted of wheat flour (200 g), yeast powder (4 g), salt (1 g) and water (to give a maximum consistency of 500 FU). The steamed bread was prepared by replacing wheat flour with 15% of oat bran (15 g/100 g *w*/*w* based on wheat flour dry weight). The dough was formed by using stand mixer (BBEK1092, Briscoes Ltd., Christchurch, New Zealand) for 5 min and kneading by hand for 5 min, then it was rested at 28 °C for 5 min. After that, the dough was kneaded for a further 8 min before fermentation in an incubator at 30 °C for 30 min. After fermentation, the dough was rolled out, and allowed to rise at 30 °C for 25 min. Finally, the dough pieces were placed in a Convotherm mini easyTouch oven (CONVOTHERM Elektrogeräte GmbH, Bavaria, Germany) and steamed for 20 min. Steamed bread loaves were cooled to room temperature and then analysed.

### 2.4. Physical Properties of Chinese Steamed Bread

Moisture content of steamed bread was determined by an oven drying method (105 ± 2 °C overnight) described by the AACC International Approved Method 44-16.01 [12]. Analysis was performed in triplicate.

The volume of steamed bread loaves was measured using the rapeseed displacement method, following the AACC International Approved Method 10-05. 01 [12]. The measurements were carried out in triplicate.

Specific volume of steamed bread was calculated by dividing loaf volume by loaf weight, according to the AACC International Approved Method 10-05. 01 [12].

The texture properties of steamed bread were determined using TA-XT2 Texture Analyser (Stable Micro Systems, Surrey, UK) equipped with a 25 mm diameter cylinder probe [13]. Steamed bread loaf was cut into slices of 25 mm thickness. The individual bread samples were compressed twice by probe to provide insight into how samples behave during chewed. The test settings were as follows: pre-test speed: 1.0 mm/s; test speed: 1.7 mm/s; post-test speed: 10.0 mm/s; strain: 40%; trigger force: 5 g. Analysis was performed in triplicate.

### 2.5. Image Analysis

Image analysis was carried out following the method described by Dewaest et al. [14] with some modifications. Briefly, a colour video camera (Sony, Digital 8 DRC-TRV-120, Tokyo, Japan) was located above the bread slices at a distance of 5 cm. The two-dimensional digital images were stored in a bit map (bmp) colour and graphics format of 24 bits, with a resolution of 640 × 480 pixels and prior to analysis. Then, images were converted to a 256 gray scale (0–255) in 8 bit format by ImageJ 1.51j8 image analysis software (National Institutes of Health, Bethesda, MD, USA). Three characteristics of the crumb were measured: the number of cells per square centimetre (cells/cm^2^), the overall mean cell area (mm^2^), the size of cell (mm).

### 2.6. Total Starch and Total, Soluble and Insoluble Dietary Fibre Analysis

The determination of total starch in the steamed bread was carried out by the Megazyme Total Starch analysis kit following the AACC standard method.

The determination of total (TDF), soluble (SDF) and insoluble (IDF) dietary fibre in the steamed bread incorporated with 15% oat bran was performed by the Megazyme Dietary Fibre analysis kit following the AACC standard method.

### 2.7. Glycaemic Response Analysis

In vitro Method Analysis: An in vitro glycaemic measurement as described by Brennan et al. [15]. Milled samples of steamed bread were treated to in vitro using 10% pepsin dissolved in 0.05 mol/L HCl for 30 min at 37 °C under constant stirring to accomplish gastric digestion. Starch digestion was continued using 0.1 mL amyloglucosidase and 5 mL of 2.5% pancreatin in 0.1 mol/L Na maleate buffer pH 6 at 37 °C for 120 min with constant mixing. Triplicate 1 mL aliquots were placed in 4 mL ethanol at 20, 60 and 120 min and reducing sugar values were measured by the 3,5-Dinitrosalicylic acid (DNS) method, using glucose references. Each sample was measured in triplicate.

### 2.8. Statistical Analysis

All data were treated by ANOVA and multiple regression analysis using Minitab 17 statistical software, version 17. 2. 1 (Minitab Pty Ltd., Sydney, Australia) at a significance level of *p* < 0.05.

## 3. Results and Discussion

### 3.1. Effect of Single Enzyme on Physicochemical and Nutritional Properties of CSB Incorporated with 15% Oat Bran

Individual effect of α-amylase, xylanase and cellulase on physicochemical properties of CSB enriched with 15% oat bran are shown in Figure 1, Figure 2, Figure 3, Figure 4, Figure 5 and Figure 6. With respect to α-amylase, the addition of α-amylase significantly (*p* < 0.05) improved the physical properties of CSB. For example, the volume, height, moisture, cohesiveness, springiness and cell size of oat bran CSB increased when adding 10 ppm α-amylase. Additionally, the addition of 10 ppm α-amylase resulted in a reduction of hardness, gumminess, chewiness and cell density. Similar results were observed by Barrera et al. [16], who reported that the addition of α-amylase significantly improved bread quality, such as increase in specific volume and decrease in crumb hardness and chewiness. Rebholz et al. [17] also indicated that addition of α-amylase significantly increased the specific volume and porosity of bread crumb due to the changes in the starch-protein network. However, there is no significant differences between oat bran CSB (control) and oat bran CSB supplemented with 10 ppm α-amylase in the chemical parameters (IDF, SDF, TDF, Total starch and AUC).

In terms of xylanase, oat bran supplemented with 70 ppm xylanase had higher value of specific volume, loaf height, moisture cohesiveness, springiness and cell size compared to the control. Moreover, xylanase addition significantly (*p* < 0.05) decreased hardness, gumminess, chewiness, springiness and cell density of oat bran CSB. Xue et al. [18] reported that the addition of xylanase to bread resulted in increasing the specific volume and decreasing firmness. Additionally, Serventi et al. [19] illustrated that adding xylanase markedly improved the loaf volume and texture of wheat-cassava bread. Moreover, Schoenlechner et al. [20] found that the addition of single xylanase significantly increased the pore area and bread volume. This observation may be due to conversion of water-unextractable AX to enzyme-solubilized AX or water-extractable AX with high molecular weight [18]. Figure 4 also shows that the SDF and TDF content slightly decreased when adding 70 ppm xylanase to oat bran bread due to the mechanism of xylanase. No significant differences were observed between control bread and oat bran bread treated by xylanase in total starch content and AUC value.

Figure 1, Figure 2, Figure 3, Figure 4, Figure 5 and Figure 6 also illustrate the effect of cellulase shows the similar trend with xylanase on the physicochemical properties of oat bran bread. According to Dahiya et al. [10], who illustrated that the presence of cellulase in bread resulted in an improvement of specific volume and crumb texture due to the hydrolysis action on the non-starch polysaccharides. Moreover, Tebben et al. [21] reported that the addition of cellulase led to an increase in specific volume and softer crumb. This observation maybe attributed to the mechanism of cellulase that hydrolyses cell wall polysaccharides [11]. These results indicate that the addition of single enzyme can improve the quality of CSB enriched with 15% oat bran.

### 3.2. Effect of Enzymes Combination on the Physicochemical and Nutritional Properties of CSB Enriched in 15% Oat Bran

The combined effects of cellulase, xylanase and α-amylase on the physicochemical properties of CSB enriched with 15% oat bran were determined using full factorial design 2^3^ (Table 1), and regression coefficients and R^2^ obtained from the full factorial design are presented in Table 2. As a result, the final empirical models for specific volume, moisture, hardness, gumminess, chewiness, cells, cell size and cell area are as follows:

W (Specific volume) = 2.32 − 0.09A − 0.04B − 0.04C + 0.01AB + 0.01AC + 0.01BC + 0.02ABC (R^2^ = 0.98)

W (Loaf height) = 58.22 − 2.02A − 0.33B − 1.01C + 0.16AC (R^2^ = 0.90)

W (Moisture) = 46.85 + 0.53A + 0.24B + 0.66C − 1.89AB + 0.43AC + 0.36BC − 0.46ABC (R^2^ = 0.99)

W (Hardness) = 313.81 + 37.20A + 21.36B − 3.45AB − 9.99AC − 24.67BC − 17.25ABC (R^2^ = 0.98)

W (Chewiness) = 311.19 + 29.63A + 7.92B + 17.61C − 7.44AB − 11.53BC − 4.09BC − 24.21ABC (R^2^ = 0.98)

W (Cells) = 48.92 + 3.83A + 1.46B − 1.96C + 4.29AB − 3.66BC (R^2^ = 0.92)

W (Cell size) = 0.72 − 0.08A − 0.04B − 0.05AB + 0.03AC + 0.08BC − 0.03ABC (R^2^ = 0.97)

W (Cell area) = 21.72 − 0.58A − 0.47B − 0.33C + 0.37AC + 0.22BC (R^2^ = 0.85)

Factors:A—α-amylase; B—xylanase; C—cellulase; AB—α-amylase*xylanase; AC—α-amylase*cellulase; BC—xylanase*cellulase; ABC—α-amylase*xylanase*cellulase.

According to Table 1, the specific volume of CSB varied between 2.19 and 2.51 when enzymes combinations were added with different concentrations. Compared to the single enzyme, the combined enzymes improved the specific volume of oat bran bread up to the highest value (2.51 mL/g) when the concentration was 6, 70 and 35 ppm. Table 2 indicates that the interaction of α-amylase, xylanase and cellulase had a positive synergistic effect on the specific volume. Similar observation was reported by Flander et al. [22], who indicated that combination of tyrosinase, laccase and xylanase significantly increased the specific volume and softness of oat bread. This observation maybe attributed to the combined degradation of β-glucan and AX by combined enzymes. Additionally, Eugenia Steffolani et al. [23] reported that combinations of glucose oxidase, α-amylase and xylanase significantly increased the specific volume and decreased firmness of bread. According to the study of Sarabhai et al. [24], addition of enzymes (glucose oxidase, xylanase and protease) significantly increased specific volume and crumb springiness while crumb hardness and cohesiveness decreased in comparison to control.

In terms of texture of oat bran CSB, the combination of α-amylase, xylanase and cellulase significantly improved the texture of CSB. As a result, there was a significant decrease in hardness and increase in springiness, cohesiveness and chewiness. However, the interaction of α-amylase, xylanase and cellulase showed a negative synergistic effect on hardness of CSB. Altuna et al. [25] illustrated that the optimum formulation of enzymes combination resulted in a lower crumb firmness of bread enriched with resistant starch than control bread due to the reverted effect of enzymes on the wheat proteins dilution. Compared to the single enzyme, the combined enzymes were more efficient in improving the texture due to the synergistic effects of enzymes. Previous research proved that two enzymes (xylanase and arabinofuranosidase) treatment had significantly greater specific volume, springiness and cohesiveness and lower crumb firmness, gumminess and chewiness than single enzyme treatment [18]. In this study, the optimum result in terms of hardness, gumminess and chewiness were observed when the enzyme concentration of α-amylase, xylanase and cellulase 6, 70 and 35 ppm, respectively.

The crumb structure of oat bran CSB was significantly influenced when enzymes combinations were added. With respect to cell density, the addition of α-amylase, xylanase and cellulase combination decreased cell density from 80.50 to 45.33 cells/cm^2^. However, there was an increase in cell size and cell area. The interaction of xylanase and cellulase shows a negative synergistic effect on cell density and a positive synergistic effect on cell size and cell density. It can be seen from Table 2 the interaction of α-amylase and cellulase shows positive synergistic effects on mean cell area and cell size, as well as a negative effect on cell density. Eugenia Steffolani et al. [23] illustrated that the mixture of glucose oxidase, α-amylase and xylanase led to a bigger crumb cells than the single enzyme. According to Ebling et al. [26], the interactive effect of amyloglucosidase, glucose oxidase and transglutaminase modified crumb structure, yielding bread with less cell density, but bigger cell area than those obtained by the treatment with singly transglutaminase due to a more open gluten network.

Table 3 and Table 4 illustrate that the effect of enzymes combination on the total starch, total, soluble and insoluble dietary fibre content and predicted glycaemic impact of oat bran CSB. As a result, the enzymes combination decreased total fibre, soluble fibre, insoluble fibre and total starch content of oat bran CSB. Similar observation was reported by Park et al. [27], who pointed that the enzyme mixtures reduced the soluble and insoluble fibre content of bread due to the enzymatic hydrolysis of enzymes. For the glycaemic response, the AUC value was varied between 318.22 and 382.20 when the combination of enzymes was added with different concentration. This observation probably due to the mechanism of enzymes and interactions of α-amylase, xylanase and cellulase. Calle et al. [28] reported that addition of protase and alcalase had a significant effect on the glycaemic index and AUC parameters of Colocasia bread. According to the study of Arte et al. [29], the addition of hydrolytic enzymes increased reducing sugar and WE-pentosan content of wheat bran. Moreover, Song et al. [30] illustrated synergistic combination of cellulase and xylanase can improve the reducing sugar concentrations of corncob, corn stover, and rice straw. Therefore, these observations can be suggested to explain the variation of glycaemic impact owing to the hydrolysis of enzymes resulting in varied reducing sugar release. Previous research has found the DF can combine with proteins and form a matrix barrier surrounding the starch granules to reduce the enzyme activity [31,32]. However, the enzymes combination can change the fibre-protein network due to the hydrolysis mechanism of α-amylase, xylanase and cellulase. The research of effect of enzyme combination on glycaemic response is limited.

## 4. Conclusions

In this study, we investigated the individual and combined effects of α-amylase, xylanase and cellulase on the physicochemical properties of CSB substituted with 15% oat bran. Compared to the single enzyme, the combined enzymes increased the specific volume and cell size to higher value and decreased hardness to lower value due to the synergistic effect of enzymes. Table 5 indicates that the optimal combination of enzymes (6, 70 and 35 ppm) can significantly improve the quality of oat bran CSB, whereas increase the reducing sugar release of CSB during 2 h in vitro digestion. For the baking industry, the consistent pursuit of high quality of product will also bring to the loss of nutritional value.

## Figures and Tables

**Figure 1 foods-12-00273-f001:**
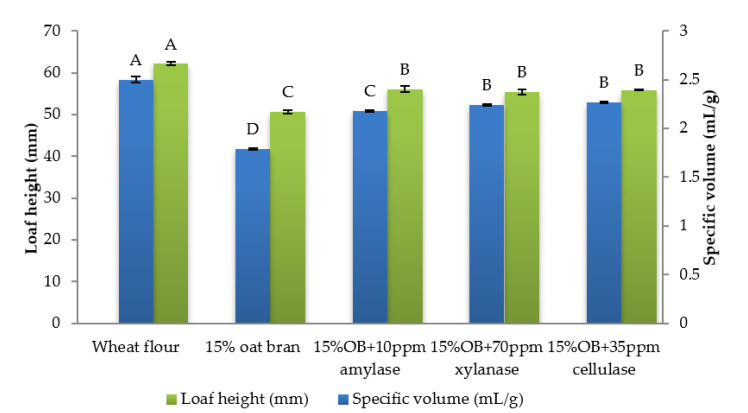
Specific volume and loaf height of Chinese steamed bread with single enzyme. Values with different letters differ significantly (*p* < 0.05).

**Figure 2 foods-12-00273-f002:**
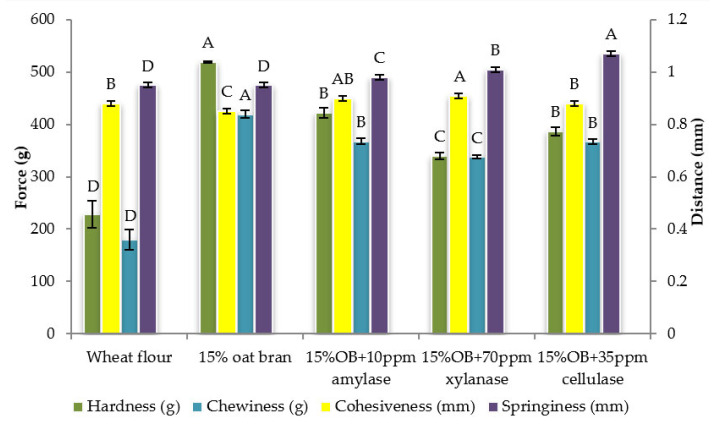
Textural properties of Chinese steamed bread with single enzyme. Values with different letters differ significantly (*p* < 0.05).

**Figure 3 foods-12-00273-f003:**
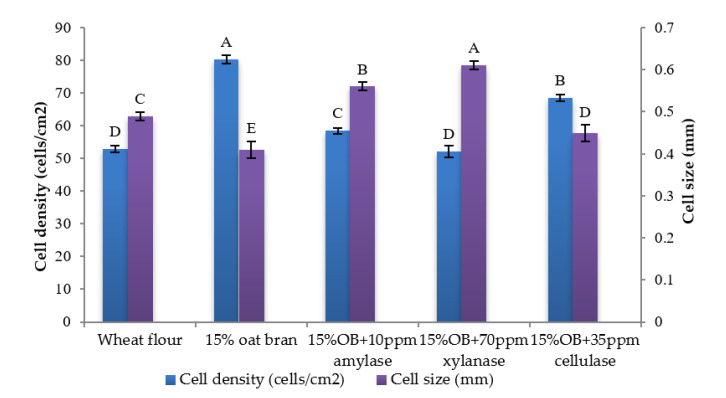
Cell density and cell size of Chinese steamed bread slices. Values with different letters differ significantly (*p* < 0.05).

**Figure 4 foods-12-00273-f004:**
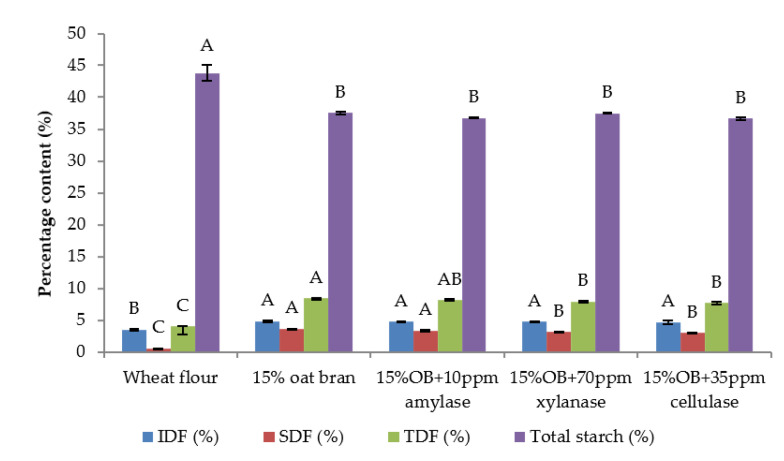
Fibre and starch content of Chinese steamed bread with single enzyme. Values with different letters differ significantly (*p* < 0.05).

**Figure 5 foods-12-00273-f005:**
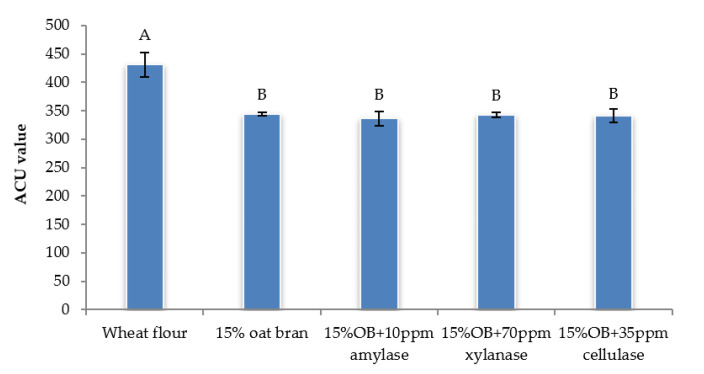
The area under curve value of Chinese steamed bread with single enzyme for predictive glycaemic response. Values with different letters differ significantly (*p* < 0.05).

**Figure 6 foods-12-00273-f006:**
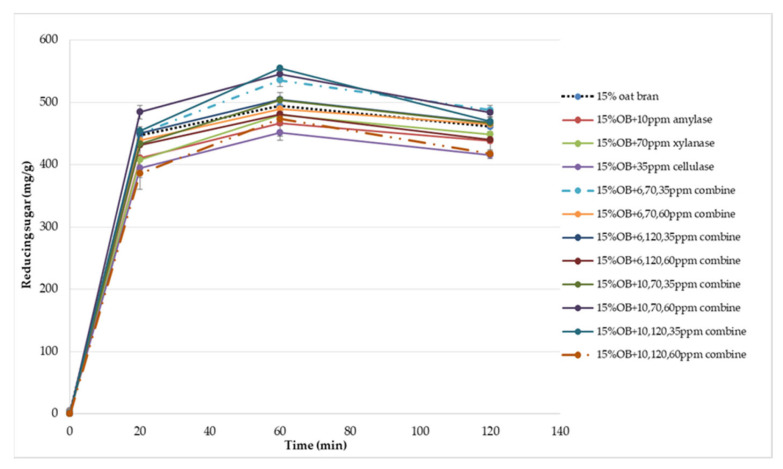
Reducing sugar released during in vitro digestion of Chinese steamed bread.

**Table 1 foods-12-00273-t001:** Effect of enzymes combination on the physical properties of CSB.

Blocks	A	B	C	Volume (mL)	Specific Volume (mL/g)	Loaf Height (mm)	Moisture (%)	Hardness (g)	Springiness (mm)	Cohesiveness (mm)	Chewiness (g)	Cells (cells/cm^2^)	Cell Size (mm)	Cell Area (%)
Wheat flour	0	0	0	248.33	2.50	62.14	40.10	228.24	0.94	0.88	179.83	53.00	0.488	21.88
Oat bran	0	0	0	194.33	1.79	50.62	45.27	519.03	0.95	0.85	419.34	80.50	0.41	20.32
1	6	70	35	266.67	2.51	60.97	44.78	233.34	1.18	0.91	257.19	46.17	0.95	23.82
2	6	70	60	265.00	2.42	60.46	43.59	270.26	1.21	0.90	275.23	49.67	0.65	21.98
3	6	120	60	252.33	2.31	59.28	49.50	305.04	1.05	0.92	346.18	38.00	0.87	21.20
4	6	120	35	260.33	2.41	60.22	47.41	297.82	1.13	0.91	247.67	46.50	0.74	22.17
5	10	70	35	251.33	2.32	58.46	47.83	300.12	1.06	0.90	305.95	45.33	0.73	21.68
6	10	120	35	242.33	2.19	57.27	44.73	419.78	1.00	0.90	363.52	65.50	0.47	20.55
7	10	120	60	246.67	2.20	54.76	46.73	318.06	1.08	0.92	319.07	51.50	0.61	21.05
8	10	70	60	239.67	2.21	54.31	50.21	366.07	0.98	0.91	374.73	48.67	0.70	21.29

All values are means. A (factor)—α-amylase; B (factor)—xylanase; C (factor)—cellulase; wheat flour—wheat flour CSB; oat bran—CSB with 15% oat bran.

**Table 2 foods-12-00273-t002:** Estimated regression coefficients of the factors of the physical properties of CSB.

Coefficient Estimate	Volume (mL)	Specific Volume (mL/g)	Loaf Height (mm)	Moisture (%)	Hardness (g)	Springiness (mm)	Cohesiveness (mm)	Chewiness (g)	Cells (cells/cm^2^)	Cell Size (mm)	Cell Area (%)
Constant (β_0_)	253.09	2.32	58.22	46.85	313.81	1.08	0.91	311.19	48.92	0.72	21.72
Amylase (β_1_)	−8.08	−0.09	−2.02	0.53	37.20	−0.06	NS	29.63	3.83	−0.88	−0.58
Xylanase (β_2_)	−2.66	−0.04	−0.33	0.24	21.36	−0.02	NS	7.92	1.46	−0.04	−0.47
Cellulase (β_3_)	−2.17	−0.04	−1.01	0.66	NS	NS	NS	17.61	−1.96	NS	−0.33
Amylase*Xylanase (β_12_)	2.18	0.01	NS	−1.89	−3.45	0.03	NS	−7.44	4.29	−0.05	NS
Amylase*Cellulase (β_13_)	NS	0.01	0.16	0.43	−9.99	NS	NS	−11.53	NS	0.03	0.37
Xylanase*Cellulase (β_23_)	NS	0.01	NS	0.36	−24.67	NS	NS	−4.09	−3.66	0.08	0.22
Amy*Xyl*Cellulase	2.74	0.02	NS	−0.46	−17.25	0.03	NS	−24.21	NS	−0.03	NS
R^2^	86.68%	98.45%	90.85%	99.58%	98.96%	90.44%	39.82%	98.19%	91.89%	96.87%	85.11%

NS—no significant effect at level (*p* < 0.05); R^2^—adjusted square coefficient (describes the percentage of variability for which the model accounts); β_0_—global means of parameters; β_1_, β_2_ and β_3_—regression coefficients corresponding to main factors; β_12_, β_13_, β_23_ and β_123_—regression coefficients corresponding to interactions; ‘−’—negative effect.

**Table 3 foods-12-00273-t003:** Effect of enzymes combination on the chemical properties of CSB.

Blocks	A	B	C	IDF %	SDF %	TDF %	Total Starch %	AUC
Wheat flour	0	0	0	3.48	0.52	4.01	43.82	491.3
Oat bran	0	0	0	4.81	3.62	8.43	37.52	344.61
1	6	70	35	4.66	3.09	7.75	34.11	371.65
2	6	70	60	4.76	2.76	7.52	35.92	346.74
3	6	120	60	4.32	3.24	7.56	33.13	336.24
4	6	120	35	4.41	3.36	7.77	34.63	355.67
5	10	70	35	4.63	3.08	7.71	35.32	351.06
6	10	120	35	4.09	3.15	7.24	36.39	375.73
7	10	120	60	4.41	2.68	7.09	33.08	318.22
8	10	70	60	4.73	2.90	7.63	34.44	382.20

All values are means. A (factor)—α-amylase; B (factor)—xylanase; C (factor)—cellulase; wheat flour—wheat flour CSB; oat bran—CSB with 15% oat bran.

**Table 4 foods-12-00273-t004:** Estimated regression coefficients of factors.

Coefficient Estimate	IDF %	SDF %	TDF %	Total Starch %	AUC
Constant (β_0_)	4.51	3.01	7.52	34.63	394.73
Amylase (β_1_)	−0.04	−0.06	−0.10	0.18	NS
Xylanase (β_2_)	−0.19	0.07	−0.12	−0.32	−8.18
Cellulase (β_3_)	0.05	−0.14	−0.09	−0.48	−8.88
Amylase*Xylanase (β_12_)	NS	−0.11	−0.14	0.25	NS
Amylase*Cellulase (β_13_)	0.06	−0.04	NS	−0.56	NS
Xylanase*Cellulase (β_23_)	NS	NS	NS	−0.71	−10.44
Amylase*Xylanase*Cellulase	0.05	−0.07	NS	NS	−11.72
R^2^	89.61%	89.36%	97.71%	96.76%	90.23%

All values are means. A (factor)—α-amylase; B (factor)—xylanase; C (factor)—cellulase; wheat flour—wheat flour CSB; oat bran—CSB with 15% oat bran.

**Table 5 foods-12-00273-t005:** Optimal solutions.

Bread Samples	Wheat Flour	15% Oat Bran (Control)	Optimum 1 67,035 ppm	Optimum 2 10,120,60 ppm
Volume (mL)	248.33 ± 2.65 B	194.33 ± 2.08 C	266.67 ± 5.18 A	246.67 ± 3.58 B
Specific volume (mL/g)	2.50 ± 0.03 A	1.79 ± 0.01 C	2.51 ± 0.02 A	2.20 ± 0.01 B
Loaf height (mm)	62.14 ± 0.38 A	50.62 ± 0.36 C	60.97 ± 1.18 A	54.76 ± 0.68 B
Moisture (%)	40.10 ± 0.01 C	45.27 ± 0.06 AB	44.78 ± 0.53 B	46.73 ± 0.62 A
Hardness (g)	228.24 ± 25.92 C	519.03 ± 1.84 A	233.61 ± 6.61 C	318.06 ± 3.62 B
Gumminess (g)	191.75 ± 19.15 C	438.80 ± 4.29 A	211.14 ± 12.02 C	292.26 ± 3.88 B
Chewiness (g)	179.83 ± 19.34 D	419.34 ± 7.34 A	257.19 ± 8.71 C	319.07 ± 3.11 B
Cohesiveness (mm)	0.88 ± 0.02 AB	0.85 ± 0.01 B	0.91 ± 0.02 A	0.92 ± 0.02 A
Springiness (mm)	0.95 ± 0.01 C	0.95 ± 0.01 C	1.18 ± 0.03 A	1.08 ± 0.02 B
Cell density (cells/cm^2^)	53.00 ± 1.03 B	80.33 ± 1.35 A	46.17 ± 3.97 C	51.50 ± 3.21 B
Cell size (mm)	0.488 ± 0.01 C	0.41 ± 0.02 D	0.95 ± 0.07 A	0.61 ± 0.01 B
Mean cell area (%)	21.88 ± 1.33 B	20.32 ± 0.76 B	23.82 ± 0.26 A	21.05 ± 0.68 B
IDF (%)	3.48 ± 0.11 C	4.81 ± 0.14 A	4.66 ± 0.01 A	4.41 ± 0.03 B
SDF (%)	0.53 ± 0.01 D	3.62 ± 0.05 A	3.09 ± 0.01 B	2.68 ± 0.01 C
TDF (%)	4.01 ± 0.10 D	8.43 ± 0.13 A	7.75 ± 0.01 C	7.09 ± 0.03 B
Total starch (%)	43.82 ± 1.30 A	37.52 ± 0.26 B	34.11 ± 0.23 C	33.08 ± 0.41 C
AUC	431.31 ± 21.4 A	344.61 ± 2.81 C	371.26 ± 8.25 B	318.22 ± 6.59 D

Means ± standard deviation (*n* = 3). Values in the same row with different letters differ significantly (*p* < 0.05).

## Data Availability

The data used to support the findings of this study are available from the corresponding author upon request.

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
