# Peer review of "Effect of Enyzmes on the Quality and Predicting Glycaemic Response of Chinese Steamed Bread"

_foods, 2023, doi:10.3390/foods12020273_

Round 1
Reviewer 1 Report
The manuscript is reviewed and find the comments are as follows
what do you mean by AUC in line 18 ?
author should mention some quantitative values of the results in the abstract.
the introduction should be concise.
What kind of images are taken is not mentioned in section 2.5.
what is the full form of DNS in line 145 ?
Table 1 is repeated. Check it.
It is easy to understand if author provide bar diagram of some of the tables.
The rationale of 15% Oat bran use is not mentioned.
The novelty of the work is not mentioned clearly. Is the combination of enzymes will help in digestion or any other properties ?
Author Response
Dear Reviewer and Editor,
Thank you for your letter and the comments concerning our manuscript. We have studied comments carefully and have made some corrections, which we hope meet with approval.
- What do you mean by AUC in line 18 ?
The area under the reducing sugar release curve (AUC) is a measurement of glycaemic response for 2 hours after Chinese steamed bread consumed. The reducing sugar release curve has been added as Figure 6.
- Author should mention some quantitative values of the results in the abstract.
Abstract has been rewritten in accordance with the comments.
- the introduction should be concise.
Introduction has been revised.
- What kind of images are taken is not mentioned in section 2.5.
The two-dimensional digital image has been mentioned in section 2.5
- what is the full form of DNS in line 145 ?
The full form of DNS is 3,5-Dinitrosalicylic acid.
- Table 1 is repeated. Check it.
Table 1 has been removed.
- It is easy to understand if author provide bar diagram of some of the tables.
Table 1 has been converted to bar graphs (Figures 1-5)
- The rationale of 15% Oat bran use is not mentioned.
The reason for using 15% oat bran has been added.
- The novelty of the work is not mentioned clearly. Is the combination of enzymes will help in digestion or any other properties ?
Abstract, introduction and conclusion have been revised. Now the novelty of this study is clear.
The manuscript has been resubmitted to your journal. We look forward to your positive response.
Kind regards
Wenjun Liu

Reviewer 2 Report
The manuscript entitled: "Synergistic effect of α-amylase, xylanase and cellulase on the physicochemical and nutritional properties of Chinese steamed bread enriched in oat bran" is about the single and synergistic effects of 3 enzymes (α-amylase, xylanase and cellulase) on characteristics of Chinese steamed bread. In general, the research is interesting and well designed. The objectives of the research are aligned with the journal's aims and scopes. There are some comments that need to address by the authors before the final decision as follows:
Title: Long title, make it short and informative.
Abstract: Rewrite it again, and improve the results. better to incorporate some quantitative data in the results.
Abstract: You need to add a general conclusion at the end of abstract.
Introduction: The problem statement is almost acceptable, improve the literature review.
Introduction: Significant of the study is not clear. Why you mixed 3 enzymes must be clarified in the introduction.
Methodology: Table 1, contains results, so it is not common to bring results in methodology. Either move it to the results part or move it to supplementary and rename it as Table S1 instead of Table 1.
Methodology: As it is a product development, better to have a sensory analysis as well.
Results and discussion: Acceptable.
Conclusion: Too long, justify your hypothesis with some future research recommendations (If any).
Author Response
Dear Reviewer and Editor,
Thank you for your letter and the comments concerning our manuscript. We have studied comments carefully and have made some corrections, which we hope meet with approval.
- Title: Long title, make it short and informative.
The title has been renamed to Effect of enyzmes on the quality and predicting glycaemic response of Chinese steamed bread.
- Abstract: Rewrite it again, and improve the results. better to incorporate some quantitative data in the results.
Abstract has been rewritten in accordance with the comments.
- Abstract: You need to add a general conclusion at the end of abstract.
Abstract has been rewritten in accordance with the comments.
- Introduction: The problem statement is almost acceptable, improve the literature review.
Introduction has been revised.
- Introduction: Significant of the study is not clear. Why you mixed 3 enzymes must be clarified in the introduction.
Introduction has been revised, and the significant of the study is clear now.
- Methodology: Table 1, contains results, so it is not common to bring results in methodology. Either move it to the results part or move it to supplementary and rename it as Table S1 instead of Table 1.
Table 1 has been removed and Table S1 has been added in Supplementary.
- Methodology: As it is a product development, better to have a sensory analysis as well.
Sensory evaluation, Molecular weight distribution and Scanning electron microscopic (SEM) will be conducted in the next paper.
- Conclusion: Too long, justify your hypothesis with some future research recommendations (If any).
Conclusion has been rewritten in accordance with the comments.
The manuscript has been resubmitted to your journal. We look forward to your positive response.
Kind regards
Wenjun Liu

Round 2
Reviewer 1 Report
The author addressed all the queries.
Reviewer 2 Report
The authors revised the manuscript fairly.